# IVAN: An Interactive Herlofson's Nomogram Visualizer for Local Weather Forecast

**Marco Angelini, Tiziana Catarci and Giuseppe Santucci \***

Department of Computer, Control, and Management Engineering Antonio Ruberti (DIAG), Sapienza University of Rome, 00185 Roma, Italy
\* Correspondence: santucci@diag.uniroma1.it

**Abstract:** In 1947, N. Herlofson proposed a modification to the 1884 Heinrich Hertz's Emagram with the goal of getting more precise hand-made weather forecasts providing larger angles between isotherms and adiabats. Since then, the Herlofson's nomogram has been used every day to visualize the results of about 800 radiosonde balloons that, twice a day, are globally released, sounding the atmosphere and reading pressure, altitude, temperature, dew point, and wind velocity. Relevant weather forecasts use such pieces of information to predict fog, cloud height, rain, thunderstorms, etc. However, despite its diffusion, non-technical people (e.g., private gliding pilots) do not use the Herlofson's nomogram because they often consider it hard to interpret and confusing. This paper copes with this problem presenting a visualization based environment that presents the Herlofson's nomogram in an easier to interpret way, allowing the selection of the right level of detail and at the same time inspection of the sounding row data and the plotted diagram. Our visual environment was compared with the classic way of representing the Herlofson's nomogram in a formal user study, demonstrating the higher efficacy and better comprehensibility of the proposed solution.

**Keywords:** information visualization; Herlofson's nomonogram; weather forecast; air sports; incremental learning

## 1. Introduction

While weather forecasts rely on models of increasing complexity (see, e.g., [1–3]) making short time local and global forecasts, climate monitoring, as well as satellite calibration and validation require the usage of local data, such as air temperature, humidity, etc. To this aim, hundreds of radiosonde balloons are released across the world, sounding the atmosphere to 10 hPa—or approximately up to 30 km altitude—and reading pressure, altitude, temperature, dew point, and wind velocity (see Figure 1a). This paper focuses on the task of local forecast for glider pilots and air sports related activities, and uses only the tropospheric part of the soundings (until about 15/16 km).

Typically, such balloons sound the atmosphere close to main airports (e.g., Milan Malpensa airport or Kirov airport) to provide detailed information useful for local forecasts (rain, thunderstorms, cloud height, fog, etc.) that can potentially affect the airport traffic security (an example of such data collected close to the Kirov airport is visible in Figure 1b). Even if the sounding data are, in principle, quite simple, i.e., few basic values collected at different altitudes, using them for forecasting even simple relevant weather events (e.g., fog, cloud height, ice, etc.) requires complex calculation based on the use of physical laws, such as the gas law, vapor saturation, heat exchange, temperature gradients, etc. As a simple example, knowing that the air temperature at ground level is 20 °C and that it is warmer than the surrounding air (thus, it will start to rise and cool down by adiabatically expanding) and that its relative humidity is 85%, two typical questions for estimating the visibility above a runway are:

1. Will the raising air generate a cloud?
2. If the answer is yes, what is the forecasted cloud base?

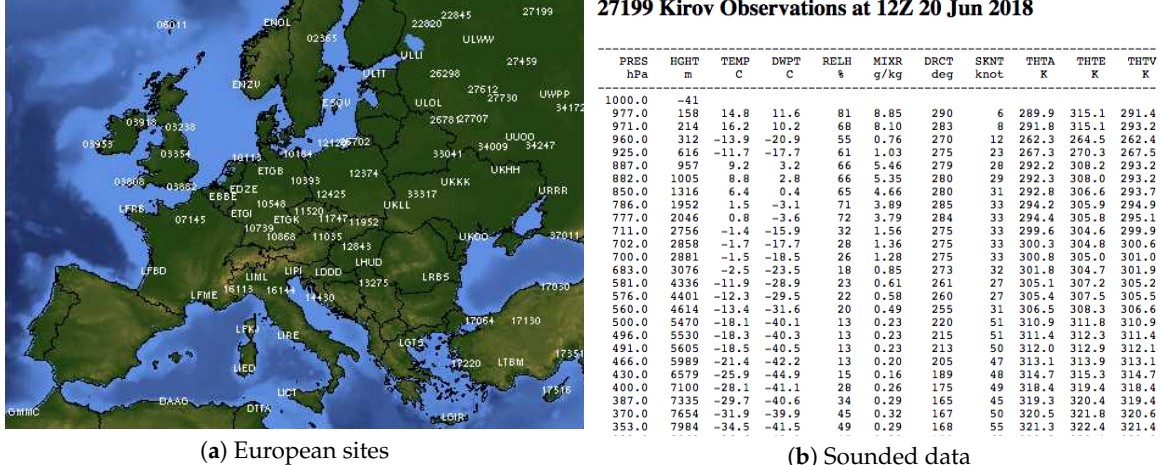

(**a**) European sites

(**b**) Sounded data

**Figure 1.** (**a**) European sites releasing radiosonde balloons twice a day (00 Zulu and 12 Zulu); and (**b**) sounding data close to Kirov airport on 20 June 2018 from ground to 7984 m.

Answering these questions requires calculating at which height the temperature of the raising air will be equal to its dew-point (i.e., the altitude at which the vapor condensation generates a cloud). This issue pushes the development of suitable nomograms plotting the relevant physical laws in order to compute *graphically* the required pieces of information: a graphical computation consists in finding the x and y values that satisfy two functions: such functions are plotted as lines on the nomogram and the solution exists on the the intersection of these two lines (the Webster definition of nomogram is:"a graphic representation that consists of several lines marked off to scale and arranged in such a way that by using a straight edge to connect known values on two lines an unknown value can be read at the point of intersection with another line"). As an example, computing the height at which the temperature of the raising air will be equal to its dew-point (i.e., the altitude at which the vapor condensation generates a cloud), requires to looking for the intersection of the curve describing the air temperature (i.e., the red state curve) with an iso-humidity curve. Nowadays, computers analyze sounding data and can easily calculate the required forecasts, without needing manual drawing of lines on a nomogram, but the Herlofson proposal is still used to show sounding data, to increase a pilot or weather expert's situational awareness about the actual situation and/or to do a rapid forecast (this is common in flight clubs, where glider pilots widely use the diagrams to forecast the strength of thermals and the height of the base of the associated cumulus clouds). Nonetheless, the inherent complexity of the concepts behind the nomogram, its crowded aspects (see Figure 2), and its differences from the classical Cartesian plane, make its comprehension and usage very difficult.

The work in [4] considers these problems by proposing an Interactive VisualizAtion enviroNment (**IVAN**) that exhibits three main features:

- It is fully configurable, allowing to explore the nomogram at different levels of complexity.
- It provides clear connection with the sounding procedure.
- It presents a high degree of interactivity, allowing to explore data details and test different hypotheses to quickly forecast weather.

This paper extends the work in [4] providing a formal evaluation of the IVAN system in the form of a user study that compared IVAN with the classical way of representing the Herlofson's nomogram, through incremental teaching activities and practical usage. The user study showed very good results with respect to its better comprehensibility and efficacy ranging from solving specific tasks to allowing data exploration.

The structure of the paper is the following: Section 2 describes related work; Section 3 introduces the main concepts underlying the nomogram; Section 4 describes the implemented prototype; Section 5 provides details and results of the users study, followed by a discussion on them. Finally, Section 6 concludes the paper and presents some future directions.

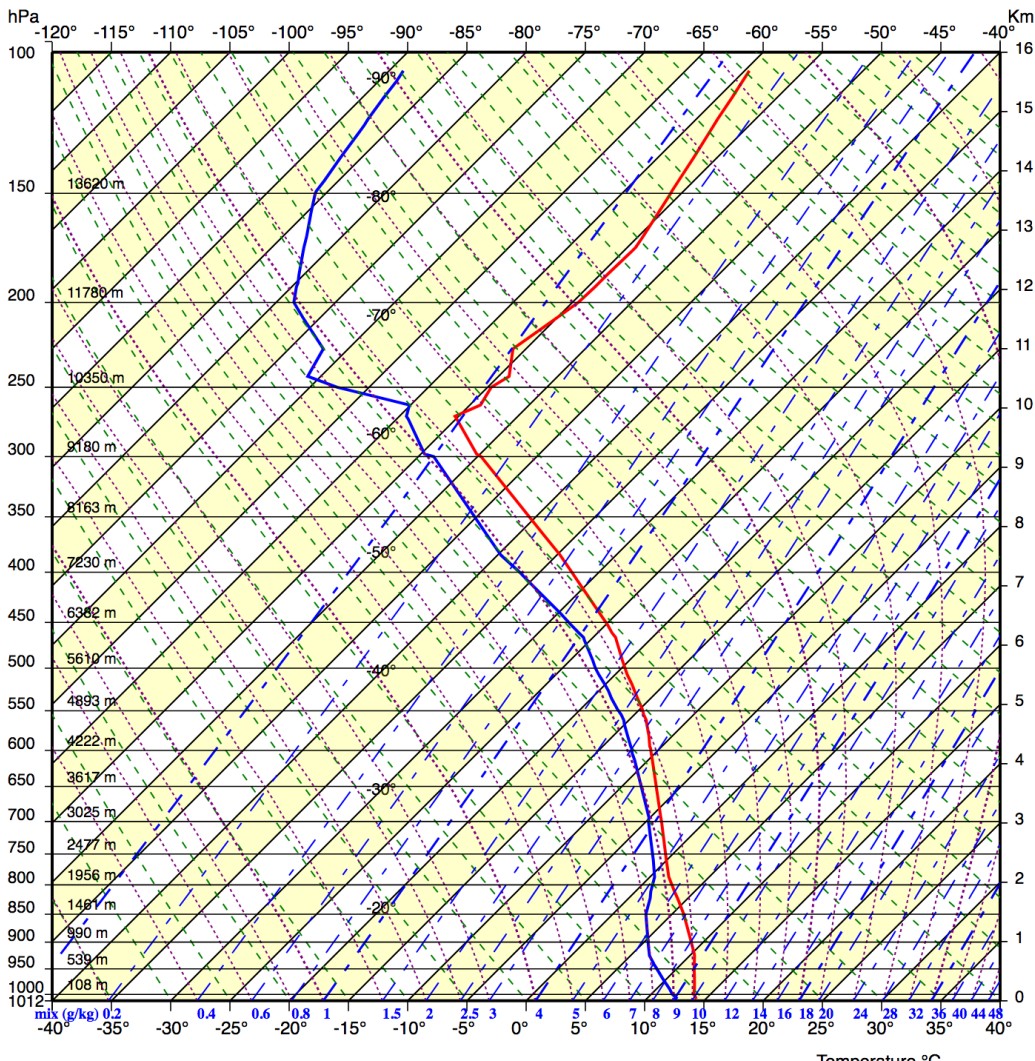

**Figure 2.** A full, crowded version of the Herlofson's nomogram that typically confuses users who are overwhelmed by lines and numbers.

## 2. Visualisation of Aerological Soundings

Information Visualization [5] as a discipline that provides useful means in order to enhance the comprehension of a phenomenon under investigation using visual representations of data, both physical and abstract. When it is coupled with the ability for a user to steer and parameterize an underlying model through the interaction with such powerful abstract data representations, it takes the name of Visual Analytics (see, e.g., [6–8]). In this context, it is very common to deal with a visual representation of mathematical concepts [9], usually under the form of diagrams. To this goal, many different plotting libraries exist that allow fast visualization of mathematical diagrams (e.g., Matplotlib and GNUplot). The problem with these libraries is the lack of interaction with the plots, allowing only a static inspection of the plotted results, and a general lack of customization other than the most general visual representations. Our proposal instead aims at providing an environment for effective

exploration and interaction with the represented diagram in order to improve learning and to enhance exploration capabilities.

Several works in information visualization tackled with the problem of weather forecast, ranging from very specific solutions (e.g., [10,11]) to more general approaches that use the weather forecast as an application domain (e.g., [12,13]). Ferstl et al. [14] proposed a new approach for analyzing the temporal growth of the uncertainty in ensembles of weather forecasts, which are started from perturbed but similar initial conditions. Lundblad et al. [15] proposed a tool for Ship and Weather Information Monitoring (SWIM) visualizing weather data combined with data from ship voyages, with the goal of monitoring a fleet and the weather development along planned routes and providing support for decisions regarding route choice and evading hazard. While a similar monitoring task is supported by our solution, the reviewed system targets only the monitoring task with no support for learning capabilities. Among the reviewed literature, very few approaches refer to the specific problem of coping with Herlofson's nomogram and not even a similar diagram for a different problem, to which our solution is targeted. The few available commercial solutions (see, e.g., [16]) are targeting expert users and do not support the configurability features of our proposed approach: additionally, they are not targeted towards incremental learning and insights discovery.

## 3. The Herlofson's Nomogram

This section has the main goal of describing the main characteristics of the Herlofson's nomogram, useful to follow the examples and the features described in Section 4. It is worth noting that N. Herlofson modified the original Heinrich Hertz's emagram (a diagram resembling an M), bending by 45° the temperature axis with the following benefits:

- increasing the angle between isotherms and adiabats cooling gradients; and
- increasing the temperature resolution with the clear advantage of getting a better precision in manually finding a crossing point between these lines (see Figure 3), generated using our interactive environment.

The nomogram deals with a multivariate problem, visualizing the following parameters (see Figure 1b): temperature, pressure, relative humidity, dew-point, and standard altitude. The discussion starts with the structure of the two axes, temperature and pressure, as shown in Figure 4a. The nomogram is indicated as skew-T, log-P because the temperature axis is rotated 45°and the pressure axis uses a logarithmic scale. Rotating the temperature axis also has the advantage of showing more values of temperature: considering Figure 3, it is also evident that the management of the temperature range of 160 °C ($-120$ °C to $+40$ °C) is done in an horizontal space able to bear a much narrow interval of 80° without compressing the X scale; a little drawback of this solution, disregarding the unfamiliar feeling of having an equation such as $x = constant$ represented by a 45° line, is that the temperature legend must be represented on both the lower and upper edges of the nomogram, because all the isotherms (but the diagonal) intersect only one of the two horizontal edges.

The second aspect to discuss is how to represent the air humidity, collected by the sounding, on a temperature/pressure plane. The chosen solution is to represent *two* temperatures for each sounding point (see Figure 4b): The first temperature corresponds to the *measured* air temperature during the sounding process, $t_1 = f(p)$, and this tabular function is interpolated by the continuous red line on Figure 4b. The second temperature corresponds to the *dew-point* temperature computed using the measured pressure and humidity, $t_2 = f(p, humidity)$, and it is interpolated by the continuous blue line. Note that, for the physical definition of dew-point, $t_2 \leq t_1$. These two status lines are called temperature and humidity curves, or equivalently state curve and dew-point curve. To discover the effective humidity of a blue point on a nomogram, a second scale is added on the bottom horizontal edge, together with dashed blue iso-humidity lines. Iso-humidity lines, visible in the nomogram in Figure 2 as dashed blue lines, represent the *maximum* humidity that can be had, at a certain combination of pressure and temperature, without the water vapor being transformed into water. That adds

confusion to the story, because iso-humidity lines are perceived as points with the same humidity while they have the same *maximum* humidity they can bear before condensing. Obviously, given a triple $< humidity, pressure, dew-point\ temperature >$, the maximum bearable humidity coincides with the actual humidity, thus iso-humidity lines can be used to read actual humidity *only* at the intersection with the blue dew-point lines.

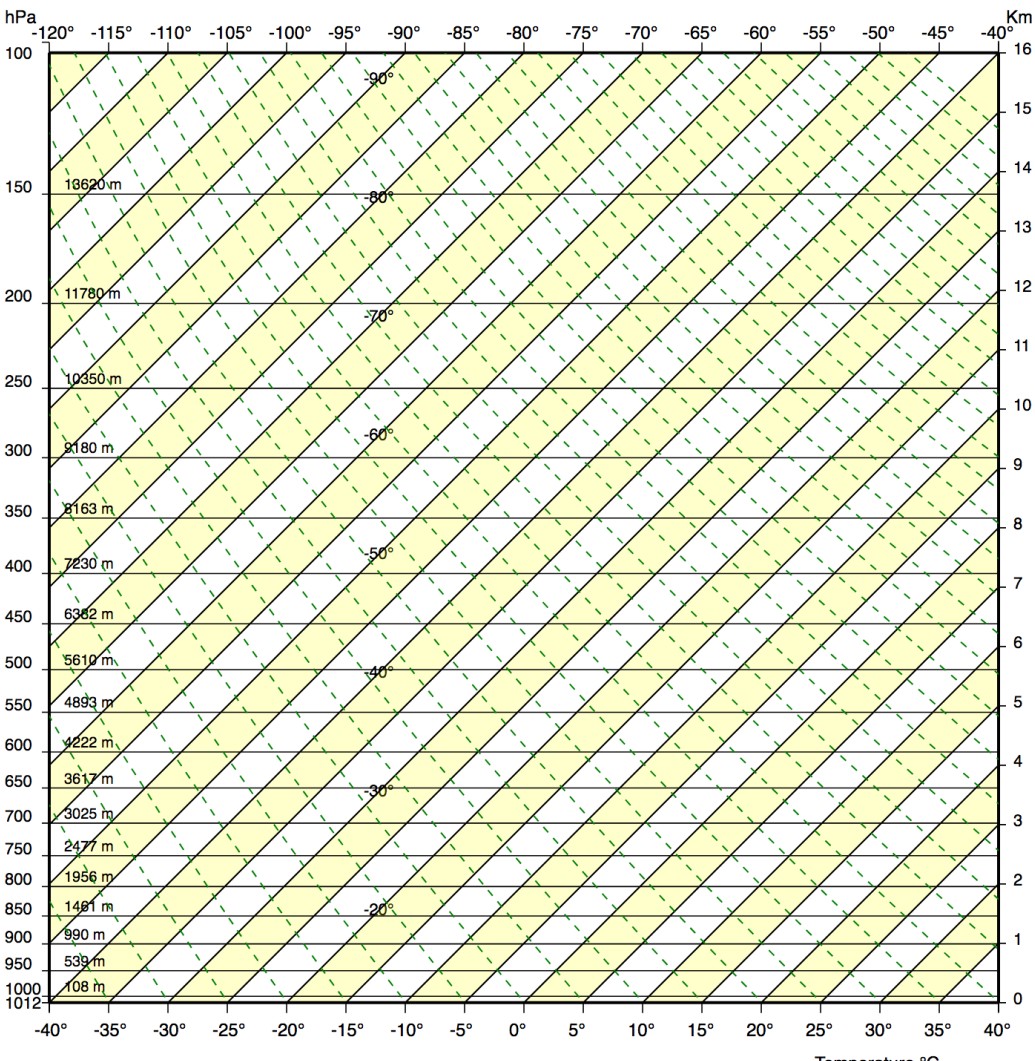

**Figure 3.** A simplified version of the Herlofson's nomogram produced by our interactive environment, displaying only isotherms (parallel to the yellow bands) and dry adiabats (green dashed lines). It is easy to grasp that the temperature axis is rotated by 45° and that isotherms exhibit a large angle (around 90°) with dry adiabats.

The previous considerations make evident the complexity that exist behind the Herlofson's nomogram and that makes hard its understanding and usage, even for a basic task such as understanding the meaning and the relationships of what is represented on it.

Having clarified the main elements of the nomogram and their meanings, we now discuss two typical usage scenarios.

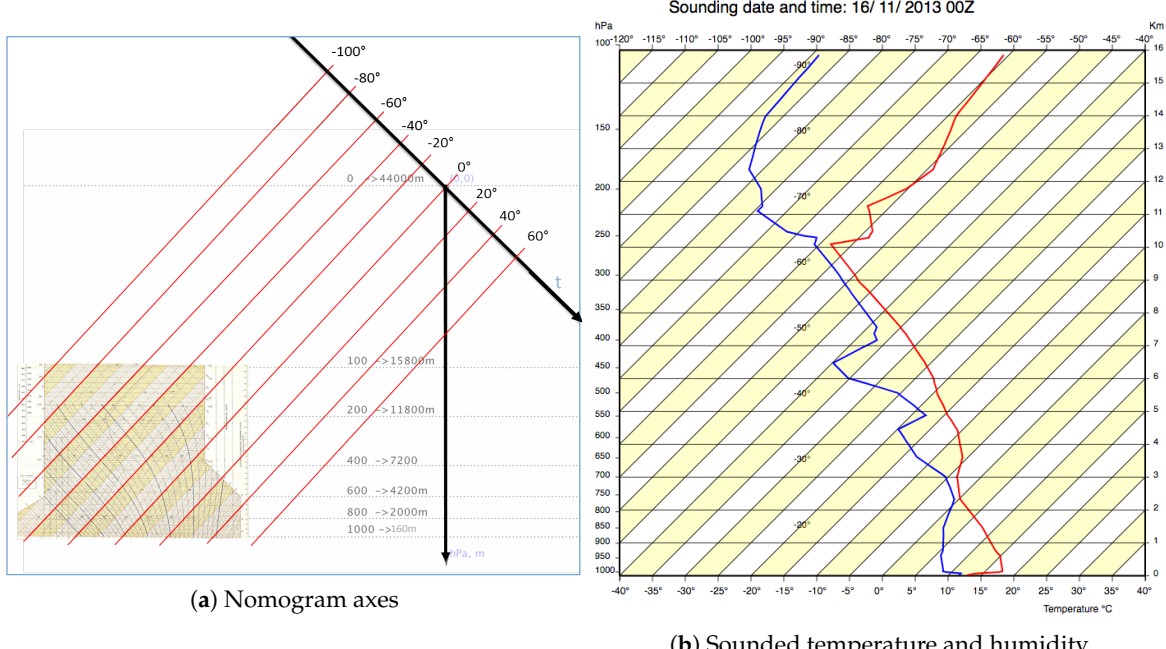

(**a**) Nomogram axes

(**b**) Sounded temperature and humidity

**Figure 4.** (**a**) The nomogram skew-T, log-P. The x-axis, temperature, is rotated 45° (skew-T), the y-axis, pressure, has an inverted scale (the higher is the pressure, the lower is the altitude) and it is logarithmic (log-P), to compensate the nonlinear relationship between pressure and altitude; the nomogram is very far from the axes origin. (**b**) The temperature state curve (red) and the humidity dew-point curve (blue).

### 3.1. Getting a Situational Awareness about the Weather Conditions

The first scenario we discuss is the usage of the nomogram to get an understanding of the weather conditions. The slope of the red line provides a first rough insight about the air stability: roughly speaking, a left slope of about 40° means that the atmosphere temperature decreases with the altitude more or less as raising air temperature does, and that implies that the strength of thermals will be constant; larger left bending calls for strong thermals, while lower bending corresponds to more stable air and weak thermals. A strong bending on the right (more than 45°) represents a temperature gradient inversion (the air temperature *increases* with the altitude) and that will stop both thermals and cloud degeneration.

Moreover, the distance between the two lines provides an indication of the relative humidity: if the lines are very close, then the actual temperature is only a little bit higher than the dew-point temperature and that implies an high likelihood of fog, clouds, and veiling of the sky.

### 3.2. Forecasting Clouds Base and Cloud Evolution

The second scenario we discuss is about forecasting cloud base and cloud evolution. Forecasting, in this case, implies making some assumptions on the near future (e.g., the temperature of the air close to the ground at 14:00 will be 24 °C) and to use other two nomogram curves: dry and wet adiabats. Such curves describe the cooling of an ascending air with (wet) or without (dry) condensed water (conversely, the warming of a descending air). Such curves are visible in Figure 4b, where dashed purple curves represent wet adiabats and dashed green curves represent dry adiabats.

In Figure 5, we see that, if the air close to the ground temperature is 25 °C, being warmer than the atmosphere temperature (red line), it climbs, cooling along a dry adiabats (thick green line), and, at an altitude of about 1.3 km and 12.5 °C (the dew-point temperature), the vapor starts to condense, generating a cloud (horizontal thick purple segment); after that point, it continues its climbing and cooling as a cloud along a wet adiabat (thick solid purple line), until crossing the red line at about

10.5 km and, having the same atmosphere temperature (about −55 °C), it stops. The generated cloud is about 9 km high (from 1.3 to 10.5 km) and there is a high risk of thunderstorms.

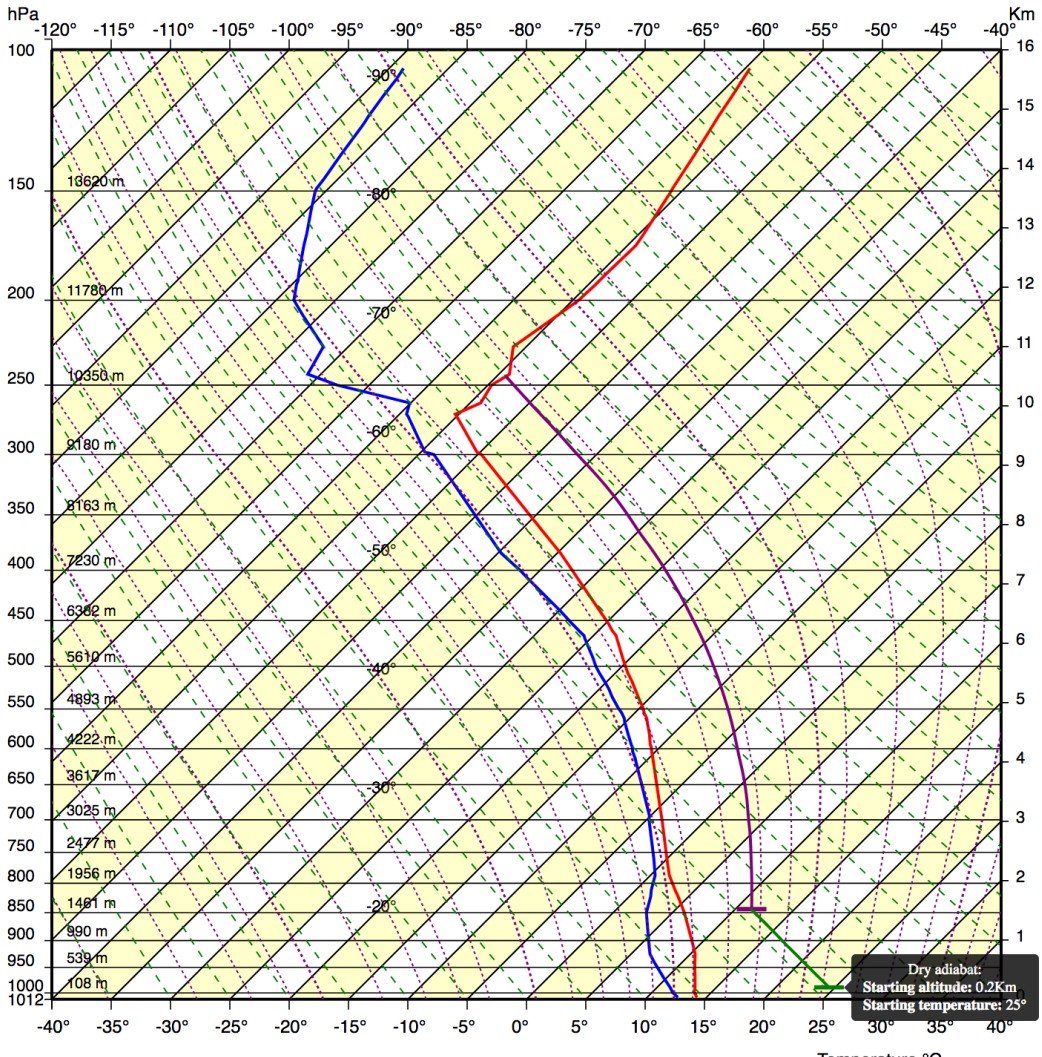

**Figure 5.** Air warmer than atmosphere rises along a dry adiabat reaching its dew-point; after that, it rises along a wet adiabats and generates a dangerous cloud that will likely produce a thunderstorm.

## 4. The IVAN System

This section describes the IVAN visual system, presenting its functionalities following an incremental approach, similar to the strategy of incremental learning presented in Section 5. IVAN is developed as a web application using Javascript and the d3.js library [17]. In this respect, IVAN constitutes an example of application of d3.js library targeted at allowing interactive exploration of mathematical charts representing physical phenomena; there are not many applications using d3.js that target this kind of scenario. Figure 6 presents an overview of the system, in which all the features of the Herlofson's nomogram are available.

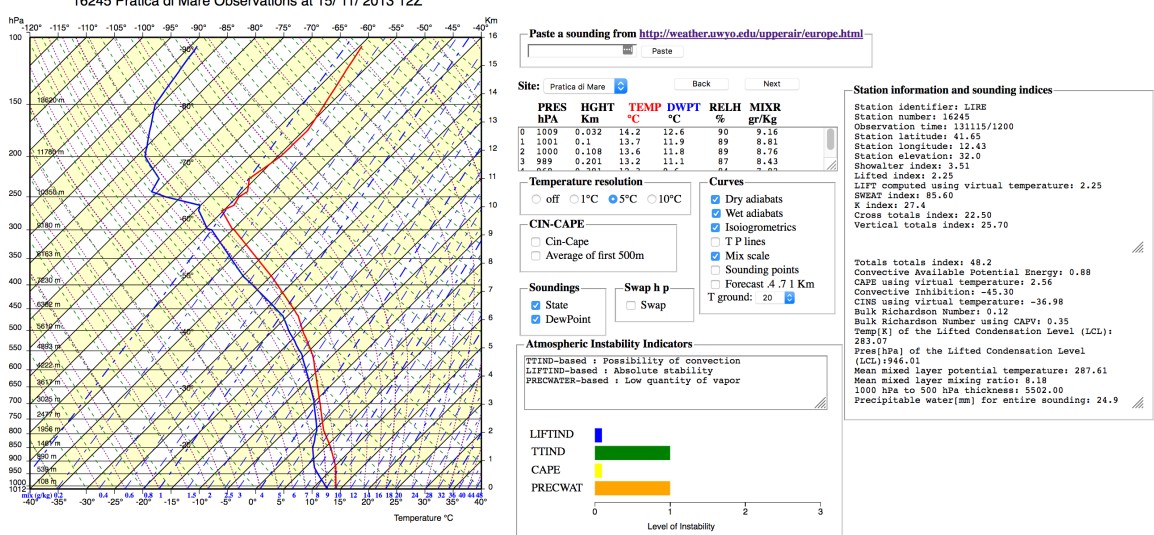

**Figure 6.** Overview of the IVAN Web application. The left side shows the Herlofson's nomogram, the central part presents detailed sounding data, configuration commands and the main atmospheric indicators, while right side reports several detailed sounding indices

On the left part, the Herlofson's nomogram is represented, allowing the user to customize its appearance with respect to the requested level of complexity. Respecting the concept of overview, it initially represents all the curves that exist in the definition of the nomogram. The user can inspect them and obtain precise values with respect to the area she is exploring. In the central part, detailed sounding data are present, in order to obtain very specific information. Additionally, commands for refining the nomogram view are present, allowing the user to set the nomogram resolution and to customize at any moment the curves to visualize, and in doing so steer the system toward the desired analysis and control at the same time overplotting, which normally affects the paper based representation of the Herlofson's nomogram. A histogram in the bottom part of the view allows reviewing indicators about air stability (e.g., CAPE). Finally, on the right is an information box, provided for expert users, which allows deepening the analysis towards additional specific information on stations and providing soundings indices.

To discuss the IVAN system behavior, we use the sequence of nine learning steps we used in our incremental learning approach strategy; steps that present the user an increasing set of pieces of information:

- Step 1: State curve, sounding points, and P, T lines are represented in this step (see Figure 7a). The system is configured to show only the state curve and the T and P axes. Moreover, sounding points are plotted on the curve, making clear that the curve is an interpolation of finite data points, whose numerical values are visible in the upper center part of Figure 6. Mouse overing a sounding point triggers the drawing of two dashed black lines that correspond to the P and the T values on the axes, helping the user in reading these values.
- Step 2: In this step, dry adiabats are added with respect to information present in Step 1, as visible in Figure 7b. The user is presented with the concept of dry adiabats and understands how to evaluate the state curve bending with respect to the dry adiabats gradient, getting the idea that, when a dry adiabat crosses the status curve, the air rising stops.
- Step 3: In this step, wet adiabats are added with respect to information presented in Step 1 (see Figure 7c). The user is presented with the concept of wet adiabats and understands how to evaluate the state curve bending with respect to the dry adiabats gradient, getting the idea that, when a wet adiabat crosses the status curve, the raising cloud stops growing.

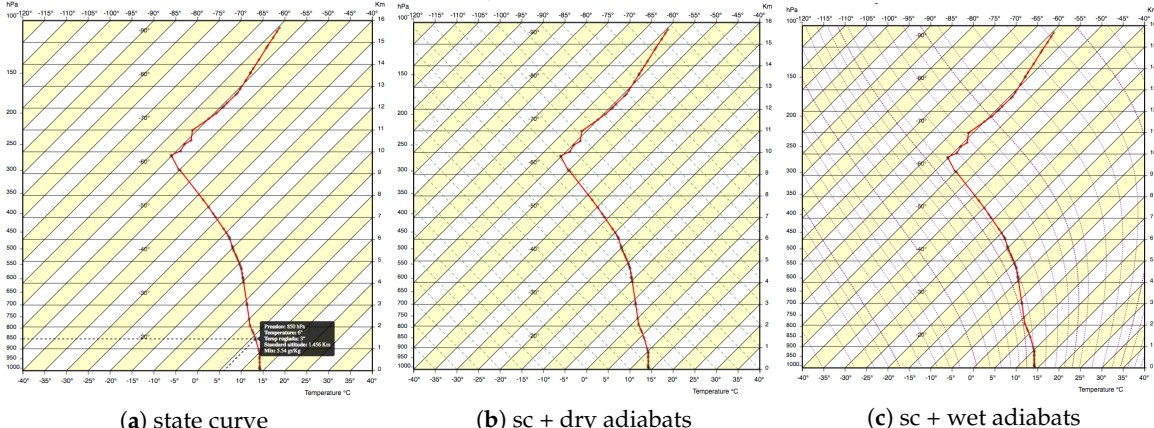

(**a**) state curve     (**b**) sc + dry adiabats     (**c**) sc + wet adiabats

**Figure 7.** (**a**) Exploring the state curve (sc), sounding points, and making explicit how to read the values of the curve; (**b**) dealing with dry adiabats; and (**c**) dealing with wet adiabats.

After the users get familiar with the basic concepts, IVAN proposes the following additional three steps:

- Step 4: State curve, dew-point curve, and P, T lines are represented in this step (see Figure 8a). The system is configured to show both the state and dew-point curves together with sounding points, making evident that both curves represent temperatures (even the "humidity" curve) and that they refer to two temperature values of the *same* sounding point. Moreover, in this way, even the distance between the curves is highlighted, explaining the risk of fog or veiling of the sky that can reduce the solar radiation.

- Step 5: In this step, the dew-point curve, mix legend and iso-humidity lines are represented (see Figure 8b). The user is presented with the definition of iso-humidity and with an additional second legend on the x-axis, and learns how to read the effective humidity values of a sounding point on the dew-point curve, using a third projection parallel to iso-humidity lines. To simplify the understanding of this operation, we remove all other lines (isotherms, red line, iso-metric, and yellow bands), reducing the overplotting and making it easier to read and use the iso-humidity lines.

- Step 6: In this step, the state curve and iso-humidity lines are represented (see Figure 8c). The confusing concept of iso-humidity is further clarified showing, through interaction, that two points on the *same* iso-humidity line, P1 and P2, have *different* humidity but share the *same* maximum humidity they can bear before condensing.

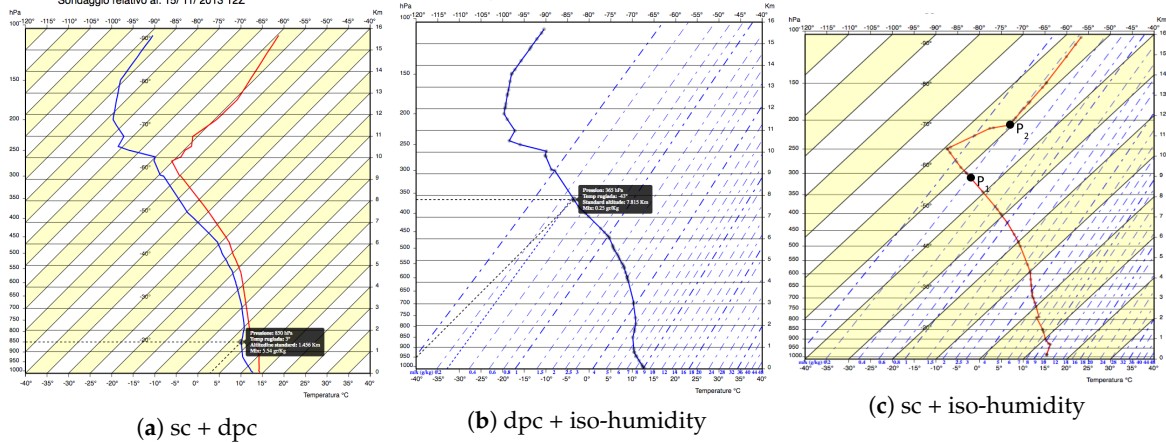

(**a**) sc + dpc     (**b**) dpc + iso-humidity     (**c**) sc + iso-humidity

**Figure 8.** (**a**) Exploring state curve (sc) and dew-point (dpc) curve; (**b**) carefully introducing the notion of iso-humidity w.r.t. dew-points curve; and (**c**) carefully introducing the notion of iso-humidity with respect to the state curve, better clarifying its meaning.

After these six steps, the user should have obtained all the basic notions needed to conduct explorative forecasting activities: with this goal, IVAN proposes these three final steps:

- Step 7: In this step, state curve, dew-point curve, iso-humidity lines, and P, T lines are represented (see Figure 9a). The system is configured to show both the state and dew-point curves together with sounding points and iso-humidity lines, and allows practicing calculating the humidity on the state curve that will be useful to forecast condensation altitudes. Selecting a sounding point on the state curve triggers selecting its corresponding point on the dew-point curve and identifying on the humidity legend the actual humidity value. That further clarifies that the two points correspond to the same sounding point and that they share P, T, humidity and dew-point temperature.

- Step 8: In this step, state curve, dry and wet adiabats and P, T lines are represented (see Figure 9b). Mouse-clicking the nomogram on the right of the state curve allows for making hypotheses on future evolution of air raising from a specific altitude at a given temperature. In the figure, the user is forecasting that the air temperature at 0.9 km altitude (e.g., the top of a little mountain) will be of 31 °C. IVAN shows that such warm air will dry climb (solid green line parallel to dry adiabats) until about 3.4 km and generate a cloud (thus, the cloud base will be 3.4 km). After that, the climbing will go on as a purple wet adiabat (parallel to wet-adiabat), generating a high cloud (thunderstorm risk) and stopping against the red state curve around 11.5 km. The user can interactively test different hypotheses, discovering situations in which the air stops before generating a cloud (blue thermals) and compare different thermals starting from different altitudes (e.g., mountain, hill, and flat land).

- Step 9: In this step, iso-humidity lines are added with respect to information present in Step 8, as visible in Figure 9c. The goal of this step is to explain how the condensation altitude can be computed on the nomogram. The user moves from the point she clicked on to the blue curve at the same altitude (0.9 km) and discovers the humidity (6.8) and the dew-point temperature (6 °C) of the rising air. Under the assumption that the composition of the air does not depend on the location but only on the altitude, he concludes that the dew-point temperature of the raising air is 6 °C: intersecting the actual dry adiabat with the 6 °C isotherm, he forecasts the condensation (and the cloud base) at about 3.4 km.

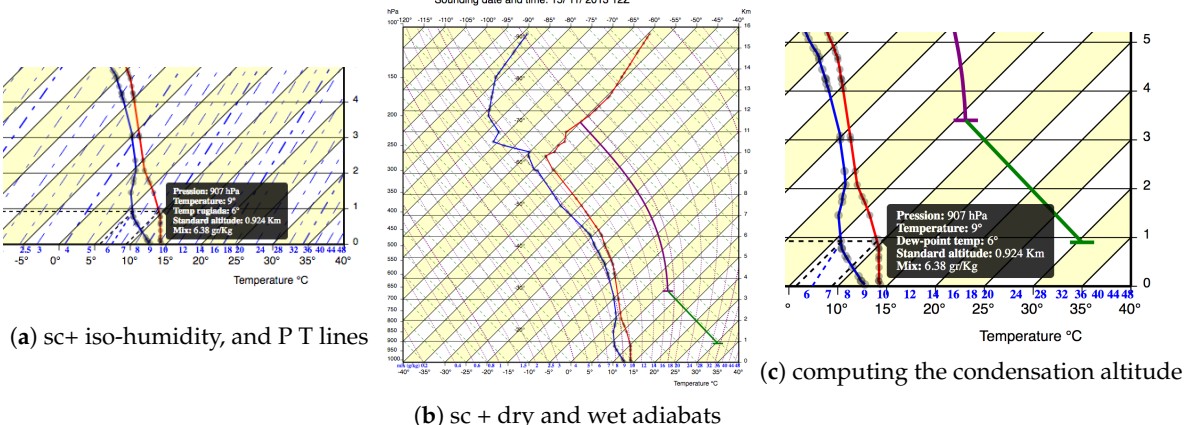

(**a**) sc+ iso-humidity, and P T lines

(**b**) sc + dry and wet adiabats

(**c**) computing the condensation altitude

**Figure 9.** (**a**) Relationship between sounding points and state and dew-point curves; (**b**) evolution of raising air along dry and wet adiabats; and (**c**) computing condensation altitude.

## 5. Evaluation

It is well known that learning is incremental and continuous; in children, this is related with the brain development process (see, e.g., [18]), and this characteristic remains also in adults. The brain forms connections with every experience we have, so that each new learning activity makes one more efficient in learning new information or acquiring new skills. In addition, presenting adult learners

with incrementally growing difficulties when getting new knowledge or performing new tasks helps them to form positive outcomes about their capabilities, motivate persistence and avoid giving them up. Providing learners opportunities to experience success in incremental steps definitely increases their self-efficacy (see, e.g., [19]). Starting from these considerations, we experimented on these ideas, comparing the IVAN system against the standard paper based Herlofson diagram in the context of teaching activity. A preliminary pilot study allowed us to tune the teaching content and diagrams image that have been used in a formal user study involving 64 users.

### 5.1. Pilot Study

We informally tested these general principles while using the IVAN system to teach to glider pilots the basic principles of the nomogram that are mandatory to correctly understand the actual weather situation and the main weather forecast (e.g., base of clouds, and risk of fog and thunderstorms). More in detail, we divided the learners into two homogeneous groups of four people. The first group was instructed with a traditional approach using paper-based diagrams of Herlofson's nomogram, while, for the second group, we adopted the interactive incremental approach implemented by IVAN and described in the previous section. At the end of the teaching period, pilot learning was assessed through a simple multiple choices test. At the end, we also administered them a satisfaction questionnaire about the learning experience. On average, members of the second group performed better in the test and gave higher marks for the quality of their experience. Results and experience gathered by this pilot study were used to design a formal controlled experiment, based on preparing an e-learning module aimed at introducing the basic principles and the graphical operation associated with the Herlofson nomogram, as detailed in the next section.

### 5.2. E-Learning Content

The e-learning module was structured on 17 slides (in Italian) covering the following topics:

- The introduction (Slides 1, 2, 3, 4, and three questions) explained the history and the purpose of the Herlofson's diagram, the structure of the x- and y-axes, the two main curves depicting sounded data, the visual indication of absolute humidity, the standard air temperature gradient, and the correct way of reading the curves. Questions asked to locate an altitude satisfying a simple criteria (e.g., the highest relative humidity) or air temperature at a given altitude.
- Cloud genesis (Slides 8, 9, and three questions) explained the temperature gradient of (wet and dry) rising air, how to compute graphically the equilibrium point, and how (and where) condensing vapor generates a cloud. Questions asked to inspect the evolution of raising till their equilibrium point (disregarding the humidity).
- Humidity (Slide 13 and two questions) showed how to use the dew-point curve to compute graphically the specific humidity. The questions asked to compute the specific humidity at a given altitude and the maximum in an interval.
- Clouds height (Slide 16 and one question) showed how to compute a cloud height generated by rising airs and considering humidity. The question asked to follow the evolution of rising air along a dry adiabat, discover its dew-point and follow the clod generation until its equilibrium, along a wet adiabat.

### 5.3. Formal User Study

We conducted a controlled experiment using as independent variable the visual representation of the nomogram: (a) the IVAN visual interface; or (b) the standard Herlofson interface. Such two interfaces were used along a 17-slide, 45-min e-learning module introducing the Herlofson diagram. The e-learning module was implemented in two versions that differ only for the diagram representations: one, which we call the Herlofson module, was based on the standard Herlofson diagram and the second one, which we call the IVAN module, was based on images coming from the

IVAN system, images that were tuned to show only the concepts the module part was introducing (see, Figures 10 and 11).

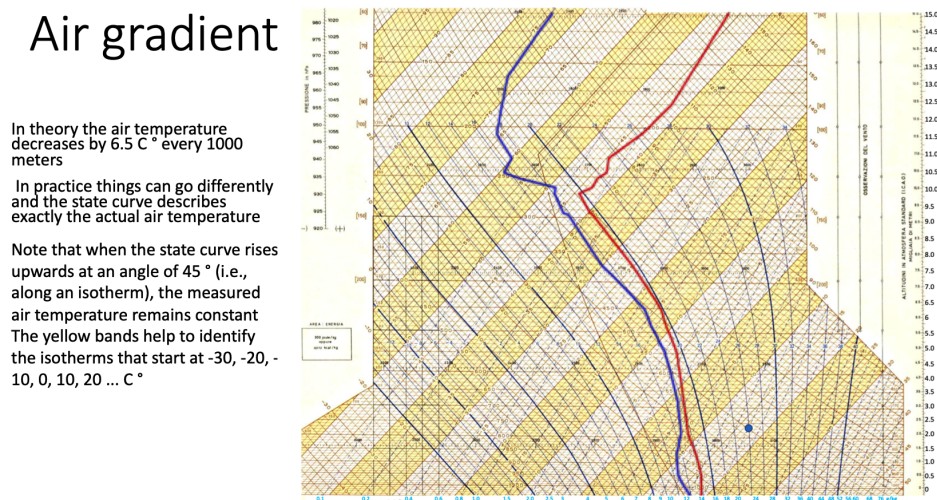

**Figure 10.** The e-learning slide (translated into English) teaching the air temperature gradient and the nomogram isotherms usage from the Herlofson module version.

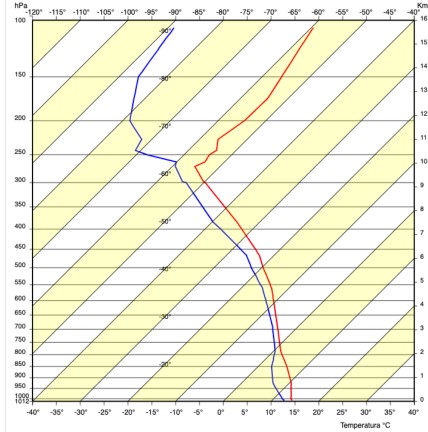

**Figure 11.** The e-learning slide (translated in English) teaching the air temperature gradient and the nomogram isotherms usage using an image from the IVAN system, in which all irrelevant concepts (i.e., dry adiabats lines, wet adiabats lines, iso-humidity lines, and humidity scale) were removed to simplify the learning activity.

The teaching activity included explanations interleaved with nine questions, aimed to assess the learning effectiveness. Answering the questions required inspecting the nomogram, locating the correct calculation line, following it on the diagram and intersecting it with the altitude or temperature line. As an example, Figure 12 (Herlofson module) and Figure 13 (IVAN module) show the question "Starting from the point indicated in the diagram and ignoring the humidity, what is the maximum altitude at which the air could rise following a dry adiabatic up to 3000 m and then a saturated adiabatic?". Answering the question required graphically following a dry adiabat until 3000 m and switching to a wet adiabat until crossing the red state curve.

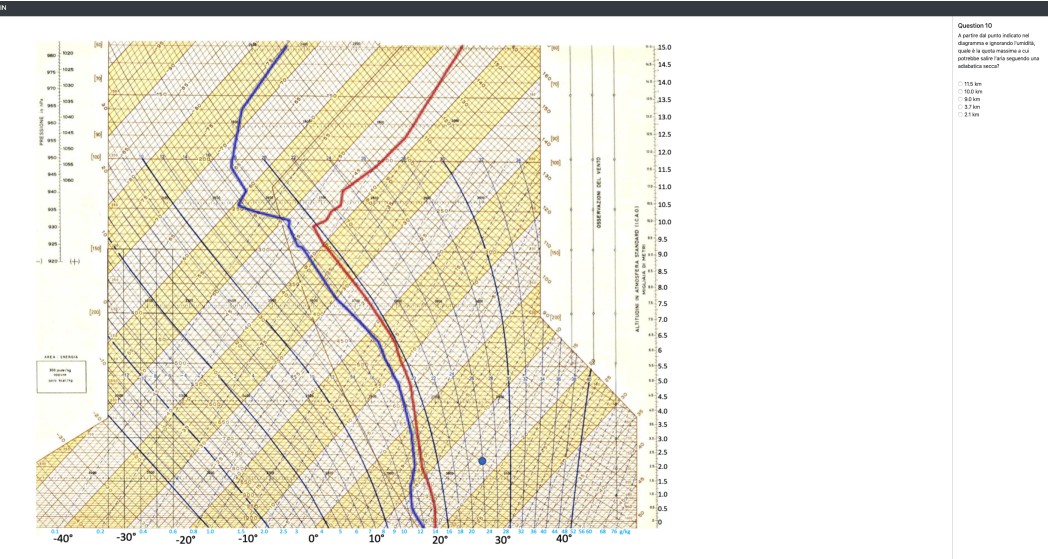

**Figure 12.** Answering a question along the e-learning activity using the Herlofson module. Images come from the STEIN system [20] that automatically collects answers, reading time, and low-level user actions.

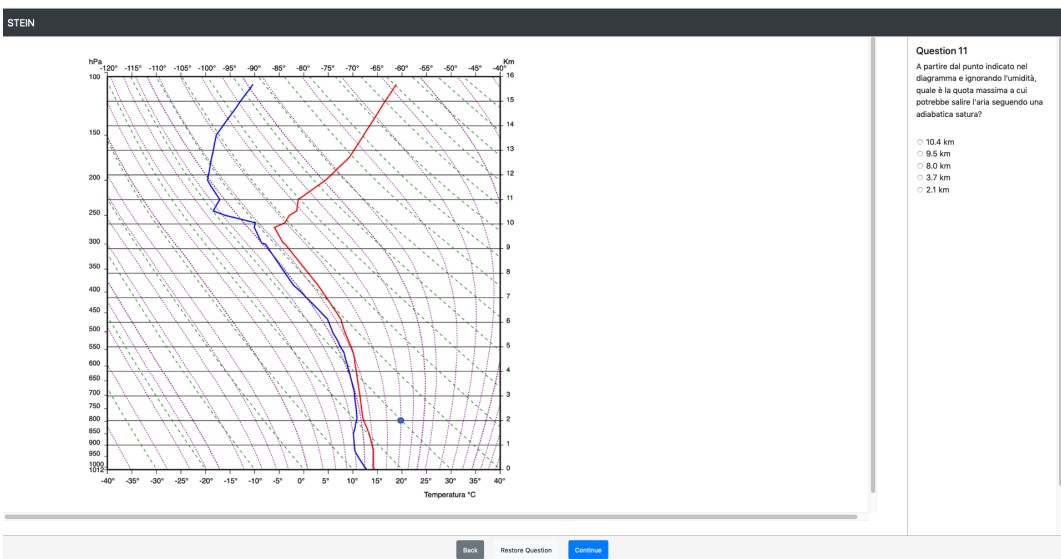

**Figure 13.** Answering the same question shown in Figure 12 using the IVAN module.

The experiment involved 64 engineering bachelor students (53 males, 11 females) ranging in age between 19 and 25 years, with no specific skills about thermodynamic diagrams but with good knowledge of ideal gas law and related physic laws about state transitions (being engineering students). Accordingly, they were considered homogeneous for the task at hand. To correctly guide the users along the experiment execution steps, i.e., reading questions, interacting with the diagrams, and reporting responses on a questionnaire, we used STEIN [20], an environment that allows for quickly integrating the system under evaluation with the user study questions, tracing her or his activities. STEIN shows a question at a time, controls the state of the system under evaluation, traces and recognizes the participants' actions, and automatically generates the response to the current question using the user input on the system, reporting it on the screen for user inspection. All user actions and elapsed times for answering to the questions were stored together with the answers, allowing for a deeper and better evaluation of the user behavior. The online version of the user study is available at http://awareserver.dis.uniroma1.it/REMS/.

**Methodology** To compare the two kinds of diagrams, people were automatically split into two groups: Group 1, 33 people (6 female, 27 male, 9 people 19 years or older) performing tasks on the Herlofson module, and Group 2, 31 people (5 female, 26 male, 10 people 19 years or older) performing the same tasks on the IVAN module. Participants were asked to read carefully the learning module content and to answer the following nine questions:

1. "At which altitude does the air have maximum relative humidity?"
2. "At what altitude the air stops being colder with increasing altitude?"
3. "What is the air temperature at 9 km?"
4. "Starting from the point indicated in the diagram and ignoring the humidity, what is the maximum altitude at which the air could rise following a dry adiabatic?"
5. "Starting from the point indicated in the diagram and ignoring the humidity, what is the maximum altitude at which the air could rise following a saturated adiabatic?"
6. "Starting from the point indicated in the diagram and ignoring the humidity, what is the maximum altitude at which the air could rise following a dry adiabatic up to 3000 m and then a saturated adiabatic?"
7. "At what altitude is the specific humidity equal to 0.8 g/kg?"
8. "Between 500 and 2000 m what is the maximum specific humidity?"
9. "$t = 20\,^{\circ}$C, altitude= 4000 m: at which altitude the rising air will generate a cloud?"

After that, participants answered the Post-Study System Usability Questionnaire (PSSUQ) [21] that consists of 16 statements scored by a seven-point Likert scale from fully disagree to fully agree. We removed five questions not applicable to the static e-learning environment (i.e., referring to error messages, online help, etc.) and we substituted the word system with the word diagram, producing the following 11 statements:

1. Overall I am satisfied with how easy it is to use this diagram.
2. It was simple to use this diagram.
3. I was able to complete the tasks and scenarios quickly using this diagram.
4. I felt comfortable using this diagram.
5. It was easy to learn to use this diagram.
6. I believe I could become productive quickly using this diagram.
7. It was easy for me to find the information I needed.
8. The information was effective in helping me complete the tasks and scenarios.
9. The organization of information on the diagram was clear.
10. I liked using the interface of this diagram.
11. Overall, I am satisfied with this diagram.

**Results** Inspecting the traces, we excluded people using more than 45 min to complete the test: some students attending the experiment perceived it as an exam and did not stop at 45 min and used all the available time to review their answers making their scores not comparable with people respecting the 45 min constraint. That reduced participants in the Herlofson group to 26 and in the IVAN group to 24. However, while considering answers to the usability questionnaire, we used all the results, being the time not relevant for that questions. From the traces, we collected four dependent variables: effectiveness (average score), time, rfficiency (score/time), and perceived usability (scores of PSSUQ). These values are reported as box plots in Figure 14. To validate the statistical significance of these figures, we performed a non-parametric test (the data were not normally distributed), i.e., a Mann–Whitney comparing ranks, which confirmed a significant effect of the independent variable "used representation of the nomogram" ($p = 0.0036$, Mann–Whitney $U = 169$) and an ANOVA test on time (the data were normally distributed), which confirmed a significant effect at the $p < 0.05$ level ($F(1.48) = 4.16$, $p = 0.0467$). Concerning the efficiency, we concluded that the user interface does *not* have a significant effect (at the $p < 0.05$ level, ($F(1.48) = 3.90$, $p = 0.053$ and $F(1.62) = 3.77$, $p = 0.056$, respectively). Concerning the PSSUQ scores, ANOVA confirmed a very significant effect of the independent variable "used representation of the nomogram" on

score at the $p < 0.05$ level ($F(1.62) = 19.73$, $p = 0.000038$). Accordingly, we can conclude that the effectiveness of people using the Herlofson module (score mean 5.80, median 6.95) is worse than effectiveness of people dealing with IVAN (score mean 6.45, median 7.73) and that completion time of people using the Herlofson module (time mean 32.23, median 34.2) is worse than completion time of people dealing with IVAN (time mean 26.6, median 25.5) On the other hand, we cannot make any assumptions on statistical differences between the two groups concerning efficiency. Concerning the perceived usability, we can conclude that the IVAN diagrams have been largely preferred with respect to traditional Herlofson diagrams.

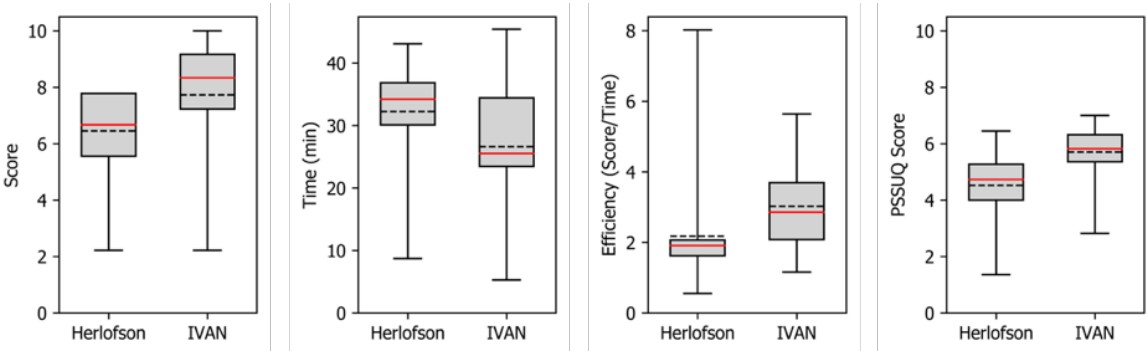

**Figure 14.** Box-plots reporting effectiveness, evaluated by the number of corrected answers normalized to 10, i.e., score, completion time, efficiency, and perceived usability. The red line represents the median, while the dotted line represents the mean. The box-plot on the left shows the distribution of the average scores obtained by counting participants' correct answers and normalizing that value to 10. The two box-plots in the middle depicts the distribution of the average time spent to complete the experiment (in min) and the ratio score/time, respectively. The box-plot on the right shows the average scores of the PSSUQ questionnaire. A between-subjects one-way ANOVA, detailed in the **Results** Section, confirmed a significant positive effect of using IVAN images on Score and Time.

## 5.4. Discussion

The statically significant difference between the average of corrected answers of people instructed using the IVAN module (score = 7.73) and that of people instructed using the Herlofson module (score = 6.45) allows us to conclude that using the IVAN visualizations produces a better understanding of the Herlofson nomogram. Moreover, the statically significant difference between the averages of time completion allows for concluding that people trained with IVAN performed faster (26.61 min) with respect to people trained with the Herlofson module (31.23 min), providing a further indication on the advantage of using IVAN visualizations. Concerning the efficiency (score/time), which does not exhibit a statistically significant difference between the two groups, we believe that IVAN completion time and, as a consequence, efficiency, was affected by the same behavior that pushed us to exclude some tests: students attending the experiment perceived it as an exam and, after having completed the test, used all the remaining 45 min to review and polish their answers. This issue is evident considering the IVAN time median (25 min) is below the mean and very close to the first quartile.

Focusing on single questions, we can conclude that, for simple questions (i.e., Q1–Q7), the number of errors are quite comparable (Figure 15), i.e., the p values table shows that a significant difference between scores exists only for Q8 and Q9. Concerning Q8, the difference in scores is very high, and both scores are the lowest in the experiment, showing that the question was very hard for both groups. However, IVAN users outperformed Herlofson users. Time was the same (no significant difference exists), likely because Herlofson users gave up without completing the question.

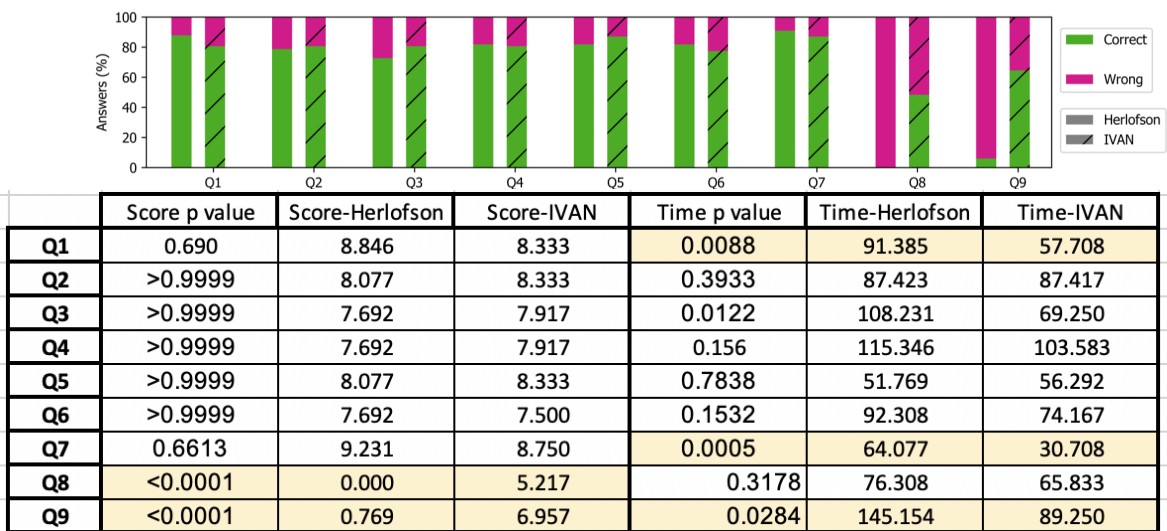

| | Score p value | Score-Herlofson | Score-IVAN | Time p value | Time-Herlofson | Time-IVAN |
|---|---|---|---|---|---|---|
| **Q1** | 0.690 | 8.846 | 8.333 | 0.0088 | 91.385 | 57.708 |
| **Q2** | >0.9999 | 8.077 | 8.333 | 0.3933 | 87.423 | 87.417 |
| **Q3** | >0.9999 | 7.692 | 7.917 | 0.0122 | 108.231 | 69.250 |
| **Q4** | >0.9999 | 7.692 | 7.917 | 0.156 | 115.346 | 103.583 |
| **Q5** | >0.9999 | 8.077 | 8.333 | 0.7838 | 51.769 | 56.292 |
| **Q6** | >0.9999 | 7.692 | 7.500 | 0.1532 | 92.308 | 74.167 |
| **Q7** | 0.6613 | 9.231 | 8.750 | 0.0005 | 64.077 | 30.708 |
| **Q8** | <0.0001 | 0.000 | 5.217 | 0.3178 | 76.308 | 65.833 |
| **Q9** | <0.0001 | 0.769 | 6.957 | 0.0284 | 145.154 | 89.250 |

**Figure 15.** Top bars: errors of users along the nine questions. It is quite evident that, for simple questions (Q1–Q7), the two groups performed in a similar way, with a percentage of errors around 20%; conversely, the most challenging questions, Q8 and Q9, requiring complex diagram navigation, exhibit a very high rate of error for the Herlofson group (errors were 100% and 94%, respectively) and a high error percentage for the IVAN group (errors were 56% and 44%, respectively). Bottom table: Means and p values of score and completion time for each question; cells highlighted in yellow represent significant differences ($p < 0.05$).

Q9 figures are quite similar to those of Q8; however, in this case, Herlofson users worked harder (the difference in time is significant), 60% more time than IVAN users, getting some correct answers. In addition, in this case, IVAN users outperformed Herlofson users.

Concerning time for Q1–Q7, we have two significant differences for Q1 and Q7. Concerning Q1, the scores are the second highest for both modules, confirming that the question was very simple. However, Herlofson users needed about 60% more time than IVAN users, reflecting the higher learning time of the Herlofson's original visualization.

Concerning Q7, the scores are the highest for both groups but Herlofson users required more than the time as IVAN users, likely because that was the first question involving iso-humidity curves.

According to the aforementioned considerations, we can conclude that IVAN outperformed the standard Herlofson diagram on the more complex questions (i.e., Q8 and Q9) that required navigating the diagram following iso-humidity and dry and wets adiabat curves. Indeed, while both groups had troubles with these questions (see Figure 15), and answering times of question Q9 show that Herlofson users spent a lot more time (145 s) than IVAN users (89 s) confirming that using the crowded Herlofson's diagram required more time to be used, the number of errors of Herlofson users increased dramatically, and none of them could get the right answer of question Q8 (compared to about 50% success of IVAN users).

Finally, the PSSUQ questionnaire confirmed that the perceived usability of IVAN is higher than the Herlofson one (average of 5.7 vs. 4.44 out of 7).

## 6. Conclusions

This paper presents an interactive visualization of the Herlofson's nomogram that has the goal of providing a better understanding of the nomogram and its usage. The high degree of interactivity and the possibility of reducing the visualized elements, considering only those relevant for the task at hand, provided the means for teaching the nomogram in an incremental way. Informal experiments of teaching and usage with glider pilots produced interesting results that were used to design a 45 min e-learning module, which was used in a formal comparative user study involving 64 people.

Results from the user study confirm the advantage of using IVAN in teaching activities. We are currently improving the interactive version of IVAN using ideas and feedback coming from the user study and we are going to design a second e-learning module incorporating interactive features and dealing with advanced usage of the Herlofson's nomogram. The up to date version of the system is available at http://awareserver.dis.uniroma1.it:8080/IVAN/ and the source code available at the following Github repository: https://github.com/aware-diag-sapienza/IVAN-normogram-visualizer.

**Author Contributions:** Conceptualization, G.S.; software, M.A.; validation, M.A., T.C., and G.S.; formal analysis, G.S.; investigation, M.A.; and writing—review and editing, M.A., T.C., and G.S.

**Funding:** This research received no external funding.

**Conflicts of Interest:** The authors declare no conflict of interest.

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
