# Peer review of "IVAN: An Interactive Herlofson’s Nomogram Visualizer for Local Weather Forecast"

_computers, doi:10.3390/computers8030053_

Round 1
Reviewer 1 Report
The present article is an extension of a work published already, as mentioned in page 2, line 56, in CEUR Workshop Proceedings - 2018 Multidisciplinary Symposium on Computer Science and ICT, REMS 2018, with the title "Visualizing the Herlofson's nomogram". Vol. 2254.
Up to section 5, the content of paper is the same of reference . Please justify that.
The new results are presented only in pages 10-13 which is basically an assessment of a questionnaire applied to users, which does not represent an innovation to Computers journal.
Despite that, they increased the number of participants to evaluate user’s experience. However, in the page 10 they mention 65 users and later there are 64 engineering bachelor students.
The text of Figures 9 and 10 are not readable. To facilitate the reading experience, authors could made available the slides as support information.
From figure 11, there is no evident advantage of proposed visualization since the error bar varies a lot in both cases.Author Response
REVIEWER 1 Open Review (x) I would not like to sign my review report ( ) I would like to sign my review report English language and style ( ) Extensive editing of English language and style required ( ) Moderate English changes required (x) English language and style are fine/minor spell check required ( ) I don't feel qualified to judge about the English language and style Yes Can be improved Must be improved Not applicable Does the introduction provide sufficient background and include all relevant references? ( ) ( ) ( ) (x) Is the research design appropriate? ( ) ( ) ( ) (x) Are the methods adequately described? ( ) ( ) ( ) (x) Are the results clearly presented? ( ) ( ) ( ) (x) Are the conclusions supported by the results? ( ) ( ) ( ) (x) Comments and Suggestions for Authors The present article is an extension of a work published already, as mentioned in page 2, line 56, in CEUR Workshop Proceedings - 2018 Multidisciplinary Symposium on Computer Science and ICT, REMS 2018, with the title "Visualizing the Herlofson's nomogram". Vol. 2254. Up to section 5, the content of paper is the same of reference . Please justify that. ANSWER: While we changed and better homogenized the content up to section 5 with respect to the conference version, the exposition of the original theory, used technique and implemented prototype was already good and changing it would have worsened the overall quality of the paper. The new results are presented only in pages 10-13 which is basically an assessment of a questionnaire applied to users, which does not represent an innovation to Computers journal. ANSWER: We expanded the evaluation section with more results, according to what reviewers asked, raising the new contributions of this journal extension. On this note, User’s evaluation are central in Visual Analytics paper, providing insights and validation to what works (and not) with the proposed solution. Despite that, they increased the number of participants to evaluate user’s experience. However, in the page 10 they mention 65 users and later there are 64 engineering bachelor students. ANSWER: Fixed The text of Figures 9 and 10 are not readable. To facilitate the reading experience, authors could made available the slides as support information. ANSWER: We have split figure 9 and 10 in two figures each, making text readable. Moreover we provide the slides of the two teaching modules translated in English as additional material. From figure 11 (figure 14 in the revised version), there is no evident advantage of proposed visualization since the error bar varies a lot in both cases. ANSWER: We have improved the caption of the figure, making evident that the differences between both score averages and time averages are statistically significant, meaning that IVAN performed better both in score and time.
Reviewer 2 Report
I thank the authors for providing a rather well-written manuscript, relevant to capacity building and training in the area of aerological soundings.
I think this point of focusing on aerology or applications for glider pilots or possibly even air sports in general (rather than analyzing atmospheric soundings up to the middle atmosphere), is important to make early in the paper, and possibly even in the title, to narrow the focus -- and possibly attract the specialized readers to this paper.
Given the nature of the problem discussed by the authors,I think it would be worth expanding figure 2 into a full-page figure.
Regarding the outline of the paper, the name section 2 (currently called "Related work") may be better reworded as "Visualisation of aerological soundings" (or similar).
Detailed comments
Introduction, 1st paragraph: the reasons for continuing to launch weather balloons is not only for "local forecasts", but includes also: global forecasts, climate monitoring, as well as satellite calibration and validation.
In fact, most soundings nowaydays aim to reach as high as possible (30 or even 10 hPa -- or approx. up to 30 km altitude), while the paper focuses only on the tropospheric part of the soundings (or approx. max. 15 km, if considering tropical locations, less at mid-latitudes).
Introduction: some of the details (such as bending of the temperature axis) would be better suited for the later sections.
Section 2: it would help to present the list of atmospheric parameters that are to be visualized, to illustrate the fact that this represents a multivariate problem.
Section 3: the list of libraries (external or not) should be better presented in a table, along with their roles/functions, so the reader understands a bit the components inside IVAN.
Section 4: What are the 9 steps intended for? Are these particular steps that the software takes the user through?
Section 5: The results of the study are too condensed (single paragraph), making it quite difficult to follow. Also, the results from the significance metrics need to be explained. I also could not find the results obtained with a 'control' group of users (taken from the same original pool of candidates, but without training).
Section 6: Rather than a server URL (which may stop at any notice), the type of software license should be presented, as well as a permanent code (or compiled library) repository.
Author Response
REVIEWER 2 Open Review (x) I would not like to sign my review report ( ) I would like to sign my review report English language and style ( ) Extensive editing of English language and style required ( ) Moderate English changes required (x) English language and style are fine/minor spell check required ( ) I don't feel qualified to judge about the English language and style Yes Can be improved Must be improved Not applicable Does the introduction provide sufficient background and include all relevant references? (x) ( ) ( ) ( ) Is the research design appropriate? (x) ( ) ( ) ( ) Are the methods adequately described? (x) ( ) ( ) ( ) Are the results clearly presented? (x) ( ) ( ) ( ) Are the conclusions supported by the results? (x) ( ) ( ) ( ) Comments and Suggestions for Authors I thank the authors for providing a rather well-written manuscript, relevant to capacity building and training in the area of aerological soundings. ANSWER: Thank you! I think this point of focusing on aerology or applications for glider pilots or possibly even air sports in general (rather than analyzing atmospheric soundings up to the middle atmosphere), is important to make early in the paper, and possibly even in the title, to narrow the focus -- and possibly attract the specialized readers to this paper. ANSWER: We have changed the title and the initial part of the introduction to better focus the paper contribution Given the nature of the problem discussed by the authors,I think it would be worth expanding figure 2 into a full-page figure. ANSWER: We have split figure 2 in two larger figures. Regarding the outline of the paper, the name section 2 (currently called "Related work") may be better reworded as "Visualisation of aerological soundings" (or similar). ANSWER: Done. Detailed comments Introduction, 1st paragraph: the reasons for continuing to launch weather balloons is not only for "local forecasts", but includes also: global forecasts, climate monitoring, as well as satellite calibration and validation. In fact, most soundings nowaydays aim to reach as high as possible (30 or even 10 hPa -- or approx. up to 30 km altitude), while the paper focuses only on the tropospheric part of the soundings (or approx. max. 15 km, if considering tropical locations, less at mid-latitudes). Introduction: some of the details (such as bending of the temperature axis) would be better suited for the later sections. ANSWER: Introduction has been modified according to the suggestion and some details have been moved in Section 2. Section 2: it would help to present the list of atmospheric parameters that are to be visualized, to illustrate the fact that this represents a multivariate problem. ANSWER: Done. Section 3: the list of libraries (external or not) should be better presented in a table, along with their roles/functions, so the reader understands a bit the components inside IVAN. ANSWER: IVAN uses only a library: d3.js. We clarified that in the text. Section 4: What are the 9 steps intended for? Are these particular steps that the software takes the user through? ANSWER: The 9 steps corresponds to the incremental learning approach. We have modified the Section makin that more clear. Section 5: The results of the study are too condensed (single paragraph), making it quite difficult to follow. Also, the results from the significance metrics need to be explained. I also could not find the results obtained with a 'control' group of users (taken from the same original pool of candidates, but without training). ANSWER: The discussion of the results and the caption and the content of the figure 14 (12 in the submission) have been extended, better explaining the conclusions that we got by the ANOVA tests. A breakdown analysis, discussing time and score for each question has been added. Concerning the experiment, it was a between-subjects one-way ANOVA test, using as independent variable the visual representation of the nomogram: a) the IVAN visual interface and b) the standard Herlofson interface; all the subjects were not aware of the nomogram by design, to not introduce a second independent variable, so there is not the notion of a ‘control’ group. We have modified the description of the experiment to make more clear the used methodology. Section 6: Rather than a server URL (which may stop at any notice), the type of software license should be presented, as well as a permanent code (or compiled library) repository. ANSWER: We added in the conclusions section a link to a public github repository that contains the source code of the prototype and all the materials with public license.
Reviewer 3 Report
This is generally a good paper. Its previous version is in the workshop and the authors extend it with a comprehensive user study. The improvement of the nomogram is good and the interactive version would definitely help the spread of nomogram. I have two suggestions for the authors to improve the paper.
First, as it is a journal of Computers, I suggest the authors introduce a bit more about visualization, information visualization, and visual analytics, by citing some more well-known introductory visualization paper or books. It would be easier for the readers to grasp the ideas of visualization.
Second, I suggest the authors discuss the concept of graphical computation of a mathematical function, which is the basis and general concept of the nomogram. It would help the readers easily understand the nomogram and its theoretical foundations.
Third, for the evaluation part, it is generally good. What I would like to see is some examples of the feedback from the participants (if the authors conduct some interviews after the experiments). By discussing representative examples, the authors can gain the idea of the users and what their suggestions are.
Author Response
REVIEWER 3 Open Review (x) I would not like to sign my review report ( ) I would like to sign my review report English language and style ( ) Extensive editing of English language and style required ( ) Moderate English changes required (x) English language and style are fine/minor spell check required ( ) I don't feel qualified to judge about the English language and style Yes Can be improved Must be improved Not applicable Does the introduction provide sufficient background and include all relevant references? ( ) (x) ( ) ( ) Is the research design appropriate? (x) ( ) ( ) ( ) Are the methods adequately described? ( ) (x) ( ) ( ) Are the results clearly presented? (x) ( ) ( ) ( ) Are the conclusions supported by the results? (x) ( ) ( ) ( ) Comments and Suggestions for Authors This is generally a good paper. Its previous version is in the workshop and the authors extend it with a comprehensive user study. The improvement of the nomogram is good and the interactive version would definitely help the spread of nomogram. I have two suggestions for the authors to improve the paper. First, as it is a journal of Computers, I suggest the authors introduce a bit more about visualization, information visualization, and visual analytics, by citing some more well-known introductory visualization paper or books. It would be easier for the readers to grasp the ideas of visualization. ANSWER: We added in the initial part of the related work section an introduction to Information visualization and Visual Analytics with references Second, I suggest the authors discuss the concept of graphical computation of a mathematical function, which is the basis and general concept of the nomogram. It would help the readers easily understand the nomogram and its theoretical foundations. ANSWER: Introduction has been extended, showing an example of a graphical computation. Third, for the evaluation part, it is generally good. What I would like to see is some examples of the feedback from the participants (if the authors conduct some interviews after the experiments). By discussing representative examples, the authors can gain the idea of the users and what their suggestions are. ANSWER: We have not conduct interviews after the experiment, so we are not able to report feedback or suggestions (the goal was not to improve the IVAN system but to assess its validity in teaching activities).
Reviewer 4 Report
The authors built a system, called IVAN, an interactive visualizer of Herlofson's Nomogram. They performed an experiments with 64 engineering students to compare their system with the existing paper based Nomogram. The results show the system has an advantage in providing comprehensibility to readers. The focus of the manuscript is enhancing the exising environment to an interactive web-based system for better interpretation. Overall, the manuscript is written in a sound and reasonable manner, and the contribution is marginal but clear. For its improvement, the following comments are given.
1. The e-learning slides shown in Figure 9 have different text formats (bullet) which should be the same for the two modules. The authors should make clear that the text contents as well as formats be the same when the subjects are in the learning process. If not, describe the difference.
2. In the experiments, the subjects were 'automatically' divided into two groups: group 1 of 33 persons and group 2 of 31 persons. After a 45-minute completion cut-off, they turned out to be 26 and 24, respectively. The authors need to show that the divided persons are randomized enough in terms of 'sex', 'age', and so forth.
3. It is unclear what the independent variable 'used diagram' was about. Please explain it.
4. Provide p-values for the 'correction rate' and 'completion time' difference in each of the 9 questions for the two groups (Herlofson and IVAN) and include them in the analysis.
5. It would be great to see Q8 and Q9 which were challenging, whether it be in the main content or in Appendix.
6. Report whether or not the conducted between-subject one-way ANOVA produced normally distributed residuals to verify the analysis model: examples are residual plots, a KS test, an Anderson-darling test, and so on.
minor comments:
- remove ')' in line 312 on page 12: '[17])'
- a e-learning -> an e-learning in 262 on page 10
Author Response
REVIEWER 4 Open Review (x) I would not like to sign my review report ( ) I would like to sign my review report English language and style ( ) Extensive editing of English language and style required (x) Moderate English changes required ( ) English language and style are fine/minor spell check required ( ) I don't feel qualified to judge about the English language and style Yes Can be improved Must be improved Not applicable Does the introduction provide sufficient background and include all relevant references? (x) ( ) ( ) ( ) Is the research design appropriate? (x) ( ) ( ) ( ) Are the methods adequately described? ( ) (x) ( ) ( ) Are the results clearly presented? ( ) (x) ( ) ( ) Are the conclusions supported by the results? ( ) (x) ( ) ( ) Comments and Suggestions for Authors The authors built a system, called IVAN, an interactive visualizer of Herlofson's Nomogram. They performed an experiments with 64 engineering students to compare their system with the existing paper based Nomogram. The results show the system has an advantage in providing comprehensibility to readers. The focus of the manuscript is enhancing the exising environment to an interactive web-based system for better interpretation. Overall, the manuscript is written in a sound and reasonable manner, and the contribution is marginal but clear. For its improvement, the following comments are given. 1. The e-learning slides shown in Figure 9 have different text formats (bullet) which should be the same for the two modules. The authors should make clear that the text contents as well as formats be the same when the subjects are in the learning process. If not, describe the difference. ANSWER: The text is the same and the missed bullet was a result of translating the original slide from Italian. We have fixed it. 2. In the experiments, the subjects were 'automatically' divided into two groups: group 1 of 33 persons and group 2 of 31 persons. After a 45-minute completion cut-off, they turned out to be 26 and 24, respectively. The authors need to show that the divided persons are randomized enough in terms of 'sex', 'age', and so forth. ANSWER: Randomization has been used for having two groups of the same cardinality while accessing the Web experiment, not to select two homogeneous groups: we have considered the 64 people homogeneous with respect to the task, all being engineering students of the first teaching year. We have clarified that in the experiment description and we have reported the compositions of the two groups. 3. It is unclear what the independent variable 'used diagram' was about. Please explain it. ANSWER: We have modified the experiment description, describing the independent variable as follows: “ We have conducted a controlled experiment using as independent variable the visual representation of the nomogram: a) the IVAN visual interface or b) the standard Herlofson interface.” 4. Provide p-values for the 'correction rate' and 'completion time' difference in each of the 9 questions for the two groups (Herlofson and IVAN) and include them in the analysis. ANSWER: We have computed the requested pvalues as an additional table in Figure 15. Such values and their implications have been discussed in the Section 5.4 Discussion. 5. It would be great to see Q8 and Q9 which were challenging, whether it be in the main content or in Appendix. ANSWER: We have provided the list of the 9 questions in the description of the experiment and the slides translated in English as additional material. 6. Report whether or not the conducted between-subject one-way ANOVA produced normally distributed residuals to verify the analysis model: examples are residual plots, a KS test, an Anderson-darling test, and so on. ANSWER: We have investigated the distribution of residuals getting a positive result (normally distributed) for time and a negative result (not normally distributed) for scores. For this reason we have used a non parametric test (Mann-Withney comparing ranks) for scores that confirmed the significance of differences. Section Results has been updated accordingly. minor comments: - remove ')' in line 312 on page 12: '[17])' - a e-learning -> an e-learning in 262 on page 10 ANSWER: Done
Round 2
Reviewer 1 Report
The authors considered all points in the revised version.
Reviewer 4 Report
The authors clarified the comments raised in the first-round review.